# GAUSS: **GrAph-customized Universal Self-Supervised Learning**

## ABSTRACT

To make Graph Neural Networks (GNNs) meet the requirements of the Web, the universality and the generalization become two important research directions. On one hand, many universal GNNs are presented for semi-supervised tasks on both homophilic and non-homophilic graphs by distinguishing homophilic and heterophilic edges with the help of labels. On the other hand, self-supervised learning (SSL) algorithms on graphs are presented by leveraging the self-supervised learning schemes from computer vision and natural language processing. Unfortunately, graph universal self-supervised learning remains resolved. Most existing SSL methods on graphs, which often employ two-layer GCN as the encoder and train the mapping functions, can't alter the low-passing filtering characteristic of GCN. Therefore, to be universal, SSL must be *customized* for the graph, i.e., learning the graph. However, learning the graph via universal GNNs is disabled in SSL, since their distinguishability on homophilic and heterophilic edges disappears without the labels. To overcome this difficulty, this paper proposes novel GrAph-customized Universal Self-Supervised Learning (GAUSS) by exploiting local attribute distribution. The main idea is to replace the global parameters with locally learnable propagation. To make the propagation matrix demonstrate the affinity between the nodes, the self-representative learning framework is employed with $k$-block diagonal regularization. Extensive experiments on synthetic and real-world datasets demonstrate its effectiveness, universality and robustness to noises.

## CCS CONCEPTS

• **Computing methodologies** → **Neural networks**;

## KEYWORDS

Graph Neural Networks, Self-supervised learning on graphs, universal representation learning, self-representative learning

**ACM Reference Format:**
Anonymous Author(s). 2018. GAUSS: GrAph-customized Universal Self-Supervised Learning. In *Proceedings of Make sure to enter the correct conference title from your rights confirmation emai (Submitted to The Web Conference '24).* ACM, New York, NY, USA, 13 pages. https://doi.org/XXXXXXX.XXXXXXX

## 1 INTRODUCTION

The graph is a general language to model non-Euclidean data. The research on graph algorithms and modeling plays a critical role in

the Web, such as the PageRank algorithm and small-world model. Most problems in graph analysis correspond to specific tasks in the Web. For example, link prediction is widely used in user/product recommendations, while community detection is employed for social mining. Recently, graph representation learning (graph embedding) has become versatile in graph analysis and has attracted much attention. Methods for graph embedding range from random walk-based models to matrix factorization-based ones and neural network-based ones.

By combining the expressive power of neural networks and the spatial [17, 24] and spectral [32] characteristics of graphs, Graph Neural Networks (GNNs) boost performance in many fields [39, 45]. Vanilla GNNs, such as GCN [12], GAT [34] and GraphSAGE [8] are designed for semi-supervised tasks, such as node classification. To make GNN meet the requirements of the Web, the universality and the generalization become two important research directions. On one hand, Web analysis requires models to be universal to both homophilic and non-homophilic graphs. For example, the citation network is homophilic, while the online dating network is heterophilic. To this end, many universal GNNs are presented for semi-supervised tasks, such as GPRGNN [4], H2GCN [47] and FAGCN [1] et al. On the other hand, training the GNNs without supervision, i.e., self-supervised on graphs, is critical, since it is difficult to obtain accurate node labels on the Web. By leveraging the self-supervised learning schemes from computer vision and natural language processing, many algorithms are introduced including DGI [35], MVGRL [9] and GCA [49].

Unfortunately, graph universal self-supervised learning remains resolved. Most existing self-supervised learning methods on graphs employ two-layer GCN as the encoder, whose essence is the low-passing filtering, and only the parameters of the mapping functions are trained. This self-supervised learning scheme is the same as in computer vision and natural language processing, and can't alter the low-passing filtering characteristic of GCN. Therefore, to be universal, self-supervised learning must be *customized* for the graph, i.e., learning the graph instead of the mapping function.

A direct and simple implementation of learning the graph in a self-supervised learning framework is the employment of universal GNNs as the encoder. The success of these universal GNNs can be ascribed to the distinguishability of the learned propagation weights of homophilic and heterophilic edges in semi-supervised learning. This is because the label information is directly or indirectly employed to learn propagation weights [10, 46]. However, this distinguishability disappears in self-supervised learning due to the lack of labels as shown in Section 3. This indicates that the existing self-supervised frameworks can't provide enough information to supervise the flexible GNN for universal representation learning.

To overcome this difficulty, this paper proposes novel GrAph-customized Universal Self-Supervised Learning (GAUSS) by exploiting local attribute distribution. The main idea is to replace the

global parameters, whose reliability is significantly weakened without the supervision of labels, with locally learnable propagation. Without the labels, a natural compromise is to make the propagation between similar nodes, which have a very high probability of belonging to the same class, and prevent the propagation between quite different nodes. That is the propagation matrix should demonstrate the affinity between the nodes. To this end, the framework of self-representative learning, which seeks the affinity matrix to represent the data itself, is employed. Furthermore, to make the learned affinity matrix possess good structure and properties, $k$-block diagonal regularization is utilized, which is defined as the sum of the $k$ smallest eigenvalues of the corresponding Laplacian matrix. To stabilize the optimization, an intermediate-term is introduced and the Alternating Direction Multiplier Method (ADMM) is used to make the subproblems strongly convex.

The main contributions of this paper are summarized as follows:

- We establish the necessity of graph-customized self-supervised learning for universal graph representation.
- We analyze the issues of employing universal GNNs as the encoder of graph self-supervised learning.
- We propose the GrAph-customized Universal Self-Supervised Learning (GAUSS) by exploiting local attribute distribution.
- We conduct extensive experiments to demonstrate its super-iMorities in high performance and robustness to noises.

## 2 NOTATIONS AND PRELIMINARIES

### 2.1 Notations

Let $\mathcal{G} = (\mathcal{V}, \mathcal{E})$ denote a graph with node set $\mathcal{V} = \{v_1, v_2, \cdots, v_N\}$ and edge set $\mathcal{E}$, where $N$ is the number of nodes. The topology of graph $\mathcal{G}$ can be represented by its adjacency matrix $\mathbf{A} = [a_{ij}] \in \{0, 1\}^{N \times N}$, where $a_{ij} = 1$ if and only if there exists an edge $e_{ij} = (v_i, v_j)$ between nodes $v_i$ and $v_j$. The degree matrix $\mathbf{D}$ is a diagonal matrix with diagonal element $d_i = \sum_{i=1}^{N} a_{ij}$ as the degree of node $v_i$. $\mathcal{N}(v_i) = \{v_j | (v_i, v_j) \in \mathcal{E}\}$ stands for the neighbourhoods of node $v_i$. Let $\mathcal{G}_i = (\mathcal{V}_i, \mathcal{E}_i)$ represents the ego-network around node $v_i$, where $\mathcal{V}_i = \mathcal{N}(v_i) \cup v_i$ and $\mathcal{E}_i$ denotes edges between nodes in $\mathcal{V}_i$. $\mathbf{X} \in \mathbb{R}^{N \times F}$ and $\mathbf{H} \in \mathbb{R}^{N \times F'}$ denote the collections of node attributes and representations with the $i^{th}$ rows, i.e., $\mathbf{x}_i \in \mathbb{R}^F$ and $\mathbf{h}_i \in \mathbb{R}^{F'}$, corresponding to node $v_i$, where $F$ and $F'$ stand for the dimensions of attribute and representation. For convenience, $\mathbf{X}_i \in \mathbb{R}^{(d_i+1) \times F}$ and $\mathbf{H}_i \in \mathbb{R}^{(d_i+1) \times F'}$ denote the collections of node attributes and representations of ego-network around node $v_i$.

### 2.2 Graph Neural Networks

Most of the Graph Neural Networks (GNNs) follow an aggregation-combination strategy [6], where each node representation is iteratively updated by aggregating node representations in the local neighbourhoods and combining the aggregated representations with the node representation itself as

$$\bar{\mathbf{h}}_v^k = \text{AGGREGATE}^k \left( \left\{ \mathbf{h}_u^{k-1} | u \in \mathcal{N}(v) \right\} \right), \qquad (1)$$

$$\mathbf{h}_v^k = \text{COMBINE}^k \left( \mathbf{h}_v^{k-1}, \bar{\mathbf{h}}_v^k \right), \qquad (2)$$

where $\bar{\mathbf{h}}_v^k$ stands for the aggregated representation from local neighbourhoods. Besides the concatenation-based implementation, such

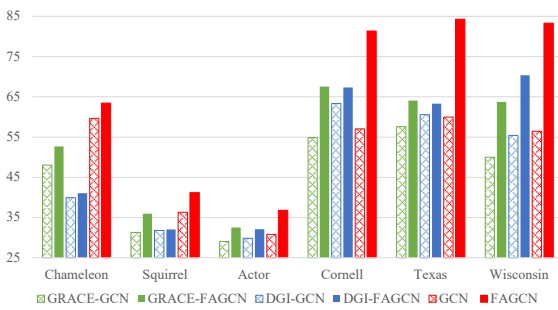

**Figure 1: Node classification performance of graph self-supervised learning with different encoders. The solid and hollow bars stand for the FAGCN and GCN as encoders, respectively. DGI [35] (blue) and GRACE [48] (green) are employed as the graph self-supervised learning framework. The red bars represent the case of semi-supervised learning.**

as GraphSAGE [8] and H2GCN [47], averaging (or summation) has been widely adopted to implement $\text{COMBINE}^k(\cdot, \cdot)$, such as GCN [12], GAT [34], GIN [42], etc. Except for the MAX and LSTM implementations in GraphSAGE [8], most of the GNNs utilize the averaging function to implement $\text{AGGREGATE}^k$. Therefore, they can be unified as

$$\mathbf{h}_v^k = \sigma \left( \left( c_{vv}^k \mathbf{h}_v^{k-1} + \sum_{u \in \mathcal{N}(v)} c_{uv}^k \mathbf{h}_u^{k-1} \right) \mathbf{W}^k \right), \qquad (3)$$

where $\mathbf{W}^k$ represents the learnable parameters and $\sigma(\cdot)$ denotes the nonlinear mapping function. Note that the scalar $c_{uv}$ is the averaging weight. For example, GCN [12] sets $c_{uv}^k = 1/(\sqrt{(d_u + 1)(d_v + 1)})$, GIN [42] sets $c_{uv}^k = 1$ for $u \neq v$ and $c_{vv}^k = 1 + \epsilon^k$, and GAT [34] learns non-negative $c_{uv}^k$ based on the attention mechanism.

**Heterophilic networks:** To handle the networks with heterophily, recent attempts make the propagation flexible by learning the propagation weights. GPRGNN [4] sets $c_{uv}^k = \gamma^k/(\sqrt{(d_u + 1)(d_v + 1)})$ with $\gamma^k$ being a learnable real value, while FAGCN [1] directly relaxes the learnable $c_{uv}$ in GAT to real value. CPGNN [46] introduces a compatibility matrix to guide the propagation by estimating the labels of all nodes with given labels, while BM-GCN [10] incorporates the block model into the GNN and learns the block structure via the given labels. Unfortunately, all these methods heavily rely on the supervision information, i.e., node labels.

### 2.3 Self-supervised Learning on Graph

Self-supervised Learning (SSL) [3] has achieved superior performance in computer vision (CV) and natural language processing (NLP). SSL on graphs [41] attempts to leverage existing SSL strategies to train the graph neural networks. SSL methods can be categorized into contrastive and predictive models [40, 41]. Predictive models train the GNNs using self-generated labels, such as topology reconstruction [11, 19, 20, 36] and property prediction [22, 26]. Unfortunately, it is difficult to determine what labels should be generated to obtain universal representations. Contrastive models conduct data augmentation/view generation and train the GNN by performing discrimination between positive pairs and negative pairs. Due to their effectiveness, universality, and simplicity, SSL

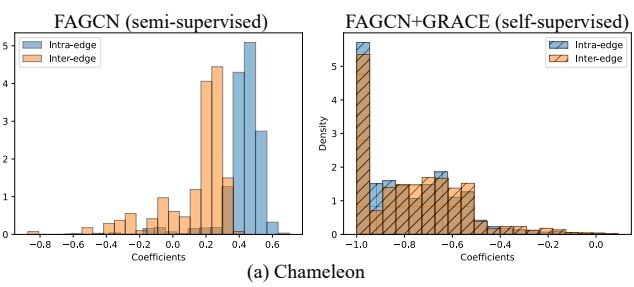

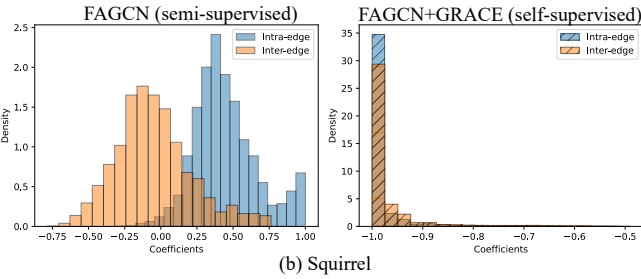

(a) Chameleon

(b) Squirrel

**Figure 2: Distributions of learned propagation weights/edge coefficients on heterophilic networks (Chameleon and Squirrel). The vanilla FAGCN and GRACE with FAGCN as encoder are representative semi-supervised and self-supervised learning methods. The blue and orange bars represent the learned weights on inter-class and intra-class edges, respectively. Note the distinguishability exits in semi-supervised learning but disappears in self-supervised one.**

on graphs pays much attention to contrastive models. Formally, contrastive models train the GNN to maximize the mutual information $\mathcal{I}(\mathbf{h}_i, \mathbf{h}_j)$ between a positive pair of node representation $\mathbf{h}_i$ and $\mathbf{h}_j$. To efficiently estimate and maximize the mutual information, two lower-bounds to the mutual information are commonly employed. InfoGraph [29], DGI [35], MVGRL [9], GMI [23] utilize the Jensen-Shannon (JS) estimator [18] and its variants as

$$\widehat{\mathcal{I}}^{(JS)}(\boldsymbol{h}_i, \boldsymbol{h}_j) = \mathbb{E}_{(\boldsymbol{A},\boldsymbol{X})\sim\mathcal{B}}\left[\log(\mathcal{D}(\boldsymbol{h}_i, \boldsymbol{h}_j))\right] + \qquad (4)$$
$$\mathbb{E}_{[(\boldsymbol{A},\boldsymbol{X}),(\boldsymbol{A}',\boldsymbol{X}')]\sim\mathcal{B}\times\mathcal{B}}\left[\log(1 - \mathcal{D}(\boldsymbol{h}_i, \boldsymbol{h}'_j))\right],$$

while GCC [25], GraphCL [44], GRACE [48] and GCA [49] use the noise-contrastive estimator (NCE) [33] as

$$\widehat{\mathcal{I}}^{(\text{NCE})}(\boldsymbol{h}_i, \boldsymbol{h}_j) = \mathbb{E}_{[(\boldsymbol{A},\boldsymbol{X}),K]\sim\mathcal{B}\times\mathcal{B}^N}\left[\log\frac{e^{\mathcal{D}(\boldsymbol{h}_i, \boldsymbol{h}_j)}}{\sum_{(\boldsymbol{A}',\boldsymbol{X}')\in K} e^{\mathcal{D}(\boldsymbol{h}_i, \boldsymbol{h}'_j)}}\right],$$

where $\mathcal{D} : \mathbb{R}^{F'} \times \mathbb{R}^{F'} \to \mathbb{R}$ is a discriminator, which is often implemented via neural networks, to determine the agreement of the two representations. $\boldsymbol{h}_i$ and $\boldsymbol{h}_j$ are the positive pair sampled from $(\boldsymbol{A}, \boldsymbol{X}) \sim \mathcal{B}$, while $\boldsymbol{h}_i$ and $\boldsymbol{h}'_j$ are the negative pair where $\boldsymbol{h}'_j$ are randomly sampled from whole graph or other graphs.

## 3 ANALYSIS

To implement universal self-supervised learning on graphs, this section analyzes the necessity of customizing SSL for graphs and the issue of employing flexible GNNs as encoders in SSL.

### 3.1 Necessity of Customizing SSL for Graph

Self-supervised learning on graphs achieves comparable performance as the semi-supervised methods on homophilic networks. Unfortunately, their performances significantly degrade on networks with heterophily. Recent research reveals that the learned representations by graph contrastive learning (GCL) essentially encode low-frequency information [14, 37]. Actually, the graph augmentations employed by GCL preserve the low-frequency information and perturb the middle- and high-frequency ones of the graph, and thus the contrastive objective tends to seek the common low-frequency information by maximizing the mutual information. This characteristic reduces the universality of the SSL on graphs.

Most graph self-supervised learning methods, especially graph contrastive learning, utilize the two-layer GCN [12] as the encoder.

Thus, the propagation scheme, i.e., $c_{uv}^k$ in Eq. (3), is fixed, and only the parameters of the mapping functions, i.e., $\mathbf{W}^k$ in Eq. (3), are trained. This is the same as self-supervised learning in computer vision and natural language processing. Actually, the essence of the GCN with a fixed propagation scheme is Laplacian smoothing [30] and low-passing filtering [38] from spatial and spectral perspectives. The training of the mapping function can't change this characteristic of GCN. Therefore, the employment of GCN leads in part to the universality limit of self-supervised learning on graphs.

### 3.2 Issue of Employing Flexible GNNs in SSL

A direct conjecture is *"Can the universality of self-supervised learning on graph be improved by the employment of universal GNNs as the encoder?"*. To answer this question, the two-layer GCN encoder in the self-supervised learning methods is replaced with FAGCN [1], which is a representative GNN to handle heterophilic networks. DGI [35] and GRACE [48], which are very different from both the objective function and data augmentation as discussed in Section 2.3, are employed as the graph self-supervised learning framework. The performances on node classification on 6 non-homophilic networks are shown in Figure 1, where the solid and hollow bars stand for the FAGCN and GCN, respectively. Experimental settings are the same as Section 5.

The differences between the solid and hollow bars with the same color stand for the performance gains by replacing the encoder. By comparing with the semi-supervised case (red bars), the performance gains in self-supervised learning (green bars for GRACE, blue bars for DGI) are limited. This indicates that the expressive power of flexible GNN encoders can't be fully exploited by self-supervised learning, and explains why most graph self-supervised learning methods use the vanilla GCN as the encoder.

The flexibility of universal GNNs is the learnable propagation scheme. Thus, to understand the behavior of flexible GNN encoder in self-supervised learning, the propagation weights learning in semi-supervised and self-supervised learning tasks are investigated and compared. To this end, the vanilla FAGCN and GRACE with FAGCN as encoder as the semi-supervised and self-supervised methods. The distributions of the learned propagation weights/edge coefficients on Chameleon and Squirrel are shown in Figure 2, where blue and orange bars represent the learned weights on inter-class and intra-class edges, respectively.

**Figure 3: The main idea of GAUSS is to replace the global parameters with locally learnable propagation.**

It can be observed that the learned propagation weights of homophilic and heterophilic edges are distinguishable in semi-supervised learning. This is because the label information is indirectly employed to train the parameters in propagation weights. CPGNN [46] and BM-GCN [10] are examples of direct exploitation of label information in learning propagation weights. However, this distinguishability disappears in self-supervised learning due to the lack of labels. This indicates that the existing self-supervised frameworks can't provide enough information to supervise the flexible GNN for universal representation learning.

## 4 METHODOLOGY

This section begins by providing the motivation for graph learning in self-supervised learning. Then, a novel graph-customized universal self-supervised (GAUSS) algorithm is proposed as long as the efficient optimization.

### 4.1 Motivations

As discussed in the previous section, existing self-supervised frameworks can't provide enough information to supervise the flexible GNNs for universal representation learning. Actually, these flexible GNNs are often parameterized with global parameters, and the flexibility for characterizing different local regions is guaranteed by the supervision information. For example, the universality of FAGCN comes from the learnable propagation weights, which are parameterized by the global parameters. The flexibility of FAGCN in capturing local homophily/heterophily characteristics is from the indirect employment of the node labels. Therefore, the reliability of global parameters is significantly weakened without the supervision of labels.

Therefore, graph-customized self-supervised learning should be locally parameterized and trained. The framework is shown in Figure 3. The propagation in each ego-network can be formulated as

$$\mathbf{H}_i = \mathbf{X}_i \mathbf{B}_i, \tag{5}$$

where $\mathbf{X}_i \in \mathbb{R}^{(d_i+1)\times F}$ and $\mathbf{H}_i \in \mathbb{R}^{(d_i+1)\times F}$ denote the collections of node attributes and representations of ego-network around node $v_i$, i.e. $\mathcal{G}_i$. $\mathbf{B}_i \in \mathbb{R}^{(d_i+1)\times(d_i+1)}$ is the propagation matrix. However, it is difficult to estimate the propagation matrix $\mathbf{B}_i$ with the help of labels as in CPGNN [46] and BM-GCN [10], since the labels of nodes are unknown in self-supervised learning tasks.

®

### 4.2 GAUSS

This subsection presents how to locally learn the propagation matrix. Note that the ideal propagation is between the nodes in the same class. Since the labels are completely unknown, a compromise

is to make the propagation between similar nodes and prevent the propagation between quite different nodes. In other words, the propagation matrix should demonstrate the affinity between the nodes. To this end, the framework of self-representative learning, which is widely used in subspace clustering [5, 13], is employed. Self-expressive learning seeks the affinity matrix, which can be used to represent data itself, i.e.

$$\mathbf{X}_i = \mathbf{X}_i \mathbf{B}_i, \quad s.t.\ \text{diag}(\mathbf{B}_i) = 0, \mathbf{B}_i \geq 0, \mathbf{B}_i = \mathbf{B}_i^{\mathsf{T}},$$

where $\text{diag}(\cdot)$ stands for the diagonal elements of the matirx, $\mathbf{B}_i \geq 0$ denotes that all elements are non-negtive, and $\mathbf{B}_i = \mathbf{B}_i^T$ represents the affinity matrix is symmetric. Thus, the objective function can be formulated as

$$\arg\min_{\mathbf{B}_i} ||\mathbf{X}_i - \mathbf{X}_i \mathbf{B}_i||^2 \tag{6}$$

$$s.t.\ \text{diag}(\mathbf{B}_i) = 0,, \mathbf{B}_i \geq 0, \mathbf{B}_i = \mathbf{B}_i^{\mathsf{T}}.$$

To make the learned affinity matrix possess good structure and properties, such as sparsity and low-rankness, some constraints are used to regularize the above learning process. Note that if the propagation is only between the nodes from the same classes, $\mathbf{B}_i$ should be a block diagonal matrix. Thus, the learned affinity matrix is excepted to be $k$-block, and Eq. (6) is enhanced to be

$$\arg\min_{\mathbf{B}_i} ||\mathbf{X}_i - \mathbf{X}_i \mathbf{B}_i||^2 + \gamma ||\mathbf{B}_i||_{kb} \tag{7}$$

$$s.t.\ \text{diag}(\mathbf{B}_i) = 0,, \mathbf{B}_i \geq 0, \mathbf{B}_i = \mathbf{B}_i^{\mathsf{T}},$$

where $||\mathbf{B}_i||_{kb}$ is the regularization to constrain $\mathbf{B}_i$ to be $k$-block diagonal, and $\gamma$ is the hyper-parameter to balance the impacts of two terms. The learned affinity matrix $\mathbf{B}_i$ can be seen as the learned new topology of the subgraph $\mathcal{G}_i$. Thus, the $k$-block diagonal $\mathbf{B}_i$ is equivalent to dividing subgraph $\mathcal{G}_i$ into $k$ connected components. The number of connected components of $\mathbf{B}_i$ is related to the spectral property of its Laplacian matrix $\mathbf{L}_{\mathbf{B}_i}$. According to [15], the following theorem holds.

**THEOREM 4.1.** *For any* $\mathbf{B} \geq 0$, $\mathbf{B} = \mathbf{P}^{\top}$, *the multiplicity* $k$ *of the eigenvalue* 0 *of the corresponding Laplacian matrix* $\mathbf{L_P}$ *equals the number of connected components (blocks) in* $\mathbf{B}$.

For any affinity matrix $\mathbf{B} \in \mathbb{R}^{n\times n}$, let $\lambda_i(\mathbf{L_P})$, $i = 1, \cdots, n-k$, be the eigenvalues of $\mathbf{L_P}$ in the decreasing order. It is known that $\mathbf{L_P} \geq 0$ and thus $\lambda_i(\mathbf{L_P}) \geq 0$ for all $i$. Then, by Theorem 4.1 , $\mathbf{B}$ has $k$ connected components if and only if

$$\lambda_i(\mathbf{L_P}) \begin{cases} > 0, & i = 1, \cdots, n-k, \\ = 0, & i = n-k+1, \cdots, n. \end{cases} \tag{8}$$

Motivated by such a property, the $k$-block diagonal regularizer can be defined as the sum of the $k$ smallest eigenvalues of $\mathbf{L_P}$, i.e,

$$||\mathbf{B}||_{kb} = \sum_{i=n-k+1}^{n} \lambda_i(\mathbf{L_P}). \tag{9}$$

It can be seen that $||\mathbf{B}||_{kb} = 0$ is equivalent to the fact that the affinity matrix $\mathbf{B}$ is $k$-block diagonal. So $||\mathbf{B}||_{kb}$ can be regarded as the block diagonal matrix structure induced regularizer.

Note that the constraints in Eq (7) may limit its representation capability and make the optimization difficult and unstable. To

**Table 1: Statistics of datasets**

| Dataset | Cora | CiteSeer | PubMed | Wiki-CS | Computers | Photo | Chameleon | Squirrel | Actor | Cornell | Texas | Wisconsin |
|---|---|---|---|---|---|---|---|---|---|---|---|---|
| #Nodes | 2,708 | 3,327 | 19,717 | 11,701 | 13,752 | 7,650 | 2,277 | 5,201 | 7,600 | 183 | 183 | 251 |
| #Edges | 5,429 | 4,732 | 44,338 | 216,123 | 245,861 | 119,081 | 36,101 | 217,073 | 33,544 | 295 | 309 | 499 |
| #Features | 1,433 | 3,703 | 500 | 300 | 767 | 745 | 2,325 | 2,089 | 932 | 1,703 | 1,703 | 1,703 |
| #Classes | 7 | 6 | 3 | 10 | 10 | 8 | 5 | 5 | 5 | 5 | 5 | 5 |

**Table 2: Results in terms of classification accuracies (in percent ± standard deviation) on homophilic benchmarks. The best and runner-up results are highlighted with bold and underline, respectively.**

| Dataset | Cora | CiteSeer | PubMed | Wiki-CS | Computers | Photo |
|---|---|---|---|---|---|---|
| GCN | 82.32±1.79 | 72.13±1.17 | 84.90±0.38 | 76.89±0.37 | 86.34±0.48 | 92.35±0.25 |
| GAT | 83.34±1.57 | 72.44±1.42 | 85.21±0.36 | 77.42±0.19 | 87.06±0.35 | 92.64±0.42 |
| MLP | 63.11±3.38 | 64.66±1.94 | 81.85±0.28 | 72.02±0.21 | 73.88±0.10 | 78.54±0.05 |
| JKNet | 79.78±1.52 | 69.80±1.76 | 85.14±0.41 | 79.52±0.21 | 85.28±0.72 | 92.68±0.13 |
| H2GCN | 83.41±1.44 | 72.19±1.18 | 85.79±0.49 | 79.73±0.13 | 84.32±0.52 | 91.86±0.27 |
| FAGCN | 82.94±3.54 | 72.38±0.80 | 86.10±0.62 | 74.34±0.53 | 83.51±1.04 | 92.72±0.22 |
| GPR-GNN | 83.89±1.66 | 72.60±1.76 | **86.79±0.56** | 79.82±0.35 | 86.71±1.82 | 92.93±0.26 |
| DeepWalk | 78.47±0.48 | 58.82±0.16 | 79.87±1.25 | 74.35±0.06 | 85.68±0.06 | 89.44±0.11 |
| node2vec | 79.24±0.90 | 59.64±0.68 | 80.47±0.86 | 71.79±0.05 | 84.39±0.08 | 89.67±0.12 |
| GAE | 76.90±0.42 | 60.22±0.43 | 82.90±0.52 | 70.15±0.01 | 85.27±0.19 | 91.62±0.13 |
| VGAE | 78.91±0.87 | 61.75±0.37 | 83.00±0.31 | 76.63±0.19 | 86.37±0.21 | 92.20±0.11 |
| DGI | 82.60±0.40 | 71.49±0.14 | 86.00±0.14 | 75.73±0.13 | 84.09±0.39 | 91.49±0.25 |
| GMI | 82.51±1.47 | 71.56±0.56 | 84.83±0.90 | 75.06±0.13 | 81.76±0.52 | 90.72±0.33 |
| MVGRL | 83.03±0.27 | 72.75±0.46 | 85.63±0.38 | 77.97±0.18 | 87.09±0.27 | 92.01±0.13 |
| GRACE | 83.30±0.40 | 71.41±0.38 | 86.70±0.34 | 79.16±0.36 | 87.21±0.44 | 92.65±0.32 |
| GCA | 82.90±0.41 | 71.21±0.24 | 86.01±0.75 | 79.35±0.12 | 87.84±0.27 | 92.78±0.17 |
| BGRL | 82.77±0.75 | 71.59±0.42 | 84.34±0.17 | 78.74±0.22 | 88.92±0.33 | 93.24±0.29 |
| GAUSS | **84.31±1.63** | **73.14±0.52** | 86.23±0.28 | **80.30±0.67** | **90.09±0.25** | **93.80±0.92** |

alleviate this issue, an intermediate term $\mathbf{Z_i}$ is introduced as follows.

$$\arg\min_{\mathbf{Z}_i, \mathbf{B}_i} \frac{1}{2}\|\mathbf{X}_i - \mathbf{X}_i\mathbf{Z}_i\|^2 + \frac{\lambda}{2}\|\mathbf{Z}_i - \mathbf{B}_i\|^2 + \gamma\|\mathbf{B}_i\|_{kb}, \quad (10)$$

$$s.t.\ \mathrm{diag}(\mathbf{B}_i) = 0, \mathbf{B}_i \geq 0, \mathbf{B}_i = \mathbf{B}_i^\top. \quad (11)$$

The optimization of Eq. (10) will be presented in the Appendix due to the limited space. Eqs (7) and (10) are equivalent when $\lambda > 0$ is sufficiently large. As will be seen in Appendix, another benefit of the term $\frac{\lambda}{2}\|\mathbf{Z}_i - \mathbf{B}_i\|^2$ is that it makes the subproblems involved in updating $\mathbf{Z}_i$ and $\mathbf{B}_i$ strongly convex and thus the solutions are unique and stable.

## 5 EVALUATIONS

In this section, the performance of our proposed GAUSS is experimentally evaluated on the node classification task. We have conducted a range of experiments to analyse and exhibit the superiority of our method in terms of its effectiveness, robustness, and visualisation.

## 5.1 Dataset

Our experiments are conducted on 12 commonly used benchmark datasets, including 6 homophilic graph datasets (i.e., Cora, CiteSeer, PubMed, Wiki-CS, Amazon Computers , and Amazon Photo [16, 27, 28]) and 6 heterophilic graph datasets (i.e., Chameleon, Squirrel,

Actor, Cornell, Texas, and Wisconsin [21]). The statistics of datasets are summarized in Table 1.

*5.1.1 Datasets and splitting.* **Cora, CiteSeer and PubMed** [27] are three citation network datasets, where nodes indicate a paper and each edge indicates a citation relationship between two papers. The labels are the research topic of papers. **Wiki-CS** [16] is a reference network constructed based on Wikipedia. The nodes correspond to articles about computer science and edges are hyperlinks between the articles. Nodes are labeled with ten classes each representing a branch of the field. **Amazon Computers and Amazon Photo** [28] are two co-purchase networks from Amazon. In these networks, each node indicates a good, and each edge indicates that two goods are frequently bought together. The labels are the category of goods. **Cornell, Texas and Wisconsin** [21] are three web page networks from computer science departments of diverse universities, where nodes are web pages and edges are hyperlinks between two web pages. The labels are types of web pages. **Chameleon and Squirrel** [21] are two Wikipedia networks where nodes denote web pages in Wikipedia and edges denote links between two pages. The labels stand for the average traffic of the web page. **Actor** [21] is an actor co-occurrence network , where nodes are actors and edges indicate two actors have co-occurrence in the same movie. The labels stand for the words of corresponding actors.

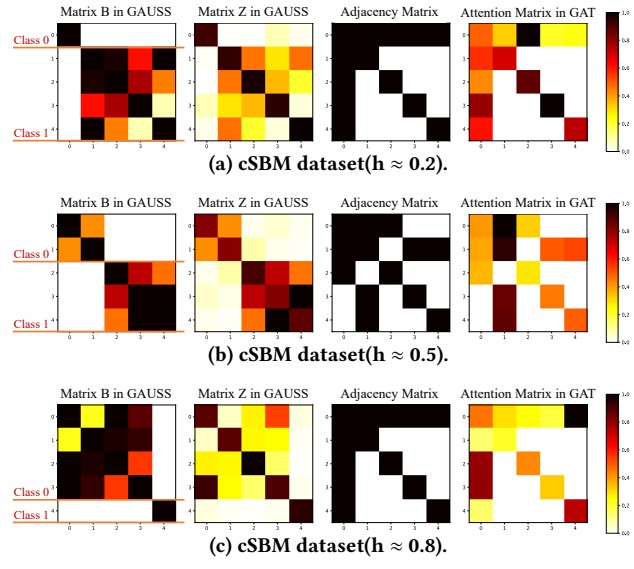

(a) cSBM dataset(h ≈ 0.2).

(b) cSBM dataset(h ≈ 0.5).

(c) cSBM dataset(h ≈ 0.8).

Figure 4: Heatmap on synthetic datasets

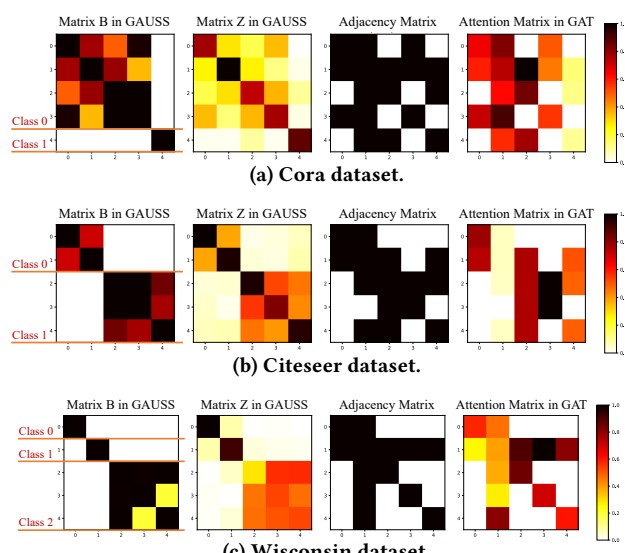

(a) Cora dataset.

(b) Citeseer dataset.

(c) Wisconsin dataset.

Figure 5: Heatmap on real world datasets

For homophilic graph datasets, we randomly split all nodes into three parts: 10% nodes for training, 10% nodes for validation and the remaining 80% nodes for testing. The performance on heterophilic graph datasets is evaluated on the commonly used 48%/32%/20% training/validation/testing.

*5.1.2 BaseLine.* To verify the superiority of the proposed GAUSS from multiple perspectives, We compare it with four groups of baseline methods: (1) The multiple layer perception (MLP) and classic GNN models for node classification task, including vanilla GCN [12] and GAT [34]; (2) GNN models designed to alleviate over-smoothing issues or networks with heterophily, including JKNet [43], GPR-GNN [4], FAGCN [1] and H2GCN [47]; (3) Conventional self-supervised graph representation learning methods, including DeepWalk [24], node2vec [7], GAE and VGAE [11]; (4) Contrastive self-supervised baselines, including DGI [35], GMI [23], MVGRL [9], GRACE [48], GCA [49] and BGRL [31].

*5.1.3 Experimental details.* All methods were implemented in Pytorch with Adam Optimizer. Some of them were implemented by a graph deep learning toolkit CoGDL [2]. We ran ten times and reported the averaged test accuracy with standard deviation. All the parameters of baselines are tuned to get a preferable performance in most situations or the same as the authors' original implementations.

*5.1.4 The GAUSS model setup and hyperparameter tuning.* There are two main parts to implementing the whole model, first using one or two layers of our module (a process that theoretically only needs to be used once for each dataset), and then the training part, which requires two or three layers of MLPs to be used for training.

In the homophily graph, we used first-order neighbours to construct the ego-network, and most of the nodes within a one-section neighbourhood belong to the same class as the central node, which is in line with the common perception. In the heterophily graph, we mainly use BFS breadth-first search to find nodes to construct ego-network, which goes beyond the limitation of first-order neighbours

to find more nodes of the same class, and also ensures that each node does not have too many nodes or too few nodes in the ego-network, which ensures the effectiveness of the iterative process. Regarding the parameters of the GAUSS process: The number of blocks k to be divided, and the parameters $\lambda$ and $\gamma$ in the iterative process, We carried out more detailed experiments on them: The other hyperparameter search space is: learning rate $\in \{0.1, 0.05, 0.01, 0.001\}$, dropout $\in \{0.2, 0.3, 0.4, 0.5, 0.8\}$. In addition, early stopping with a patience of 200 epochs and L2 regularization with coefficient $\in \{1e-2, 5e-3, 1e-3, 5e-4\}$ are used to avoid overfitting.

## 5.2 Experimental Results

*5.2.1 Results analysis on node classification.* The mean classification accuracy with a standard deviation of 6 homophilic datasets and 6 heterophilic datasets are presented in Table 2 and Table 3, respectively. We compare the proposed GAUSS with the baselines. First of all, we observe that GAUSS outperforms all baseline methods in 11 out of 12 benchmarks.

We constructed the ego-network based on the original topology, re-learns the relationships between nodes through iteration, which can effectively capture the connection of nodes with similar attributes. By adding the block diagonal representation to the iteration process, we limit the propagation between blocks with dissimilar attributes, thus achieving better performance.

We find that GAUSS significantly outperforms conventional and contrastive methods. In particular, equipping contrastive methods with heterophily-aware encoders (e.g., GRACE-FA) yields only a minor performance gain, suggesting that heterophilic graphs need crafted designs rather than simply modifying the encoder. Furthermore, when compared to supervised approaches such as GPR-GNN, FAGNN and H2GCN, which are all GNNs designed to process heterophilic datasets, we observe that GAUSS achieves new state-of-the-art results on all heterophilic datasets. These results suggest that our proposed GAUSS is more effective and universal than the previous models in processing datasets with both homophily and heterophily for node classification.

**Table 3: Results in terms of classification accuracies (in percent ± standard deviation) on heterophilic benchmarks. The best and runner-up results are highlighted with bold and underline, respectively.**

| Methods | Chameleon | Squirrel | Actor | Cornell | Texas | Wisconsin |
|---|---|---|---|---|---|---|
| GCN | 59.63±2.32 | 36.28±1.52 | 30.83±0.77 | 57.03±3.30 | 60.00±4.80 | 56.47±6.55 |
| GAT | 56.38±2.19 | 32.09±3.27 | 28.06±1.48 | 59.46±3.63 | 61.62±3.78 | 54.71±6.87 |
| MLP | 46.91±2.15 | 29.28±1.33 | 35.66±0.94 | 81.08±7.93 | 81.62±5.51 | 84.31±3.40 |
| JKNet | 58.31±2.76 | 42.24±2.11 | 36.47±0.51 | 56.49±3.22 | 65.35±4.68 | 51.37±3.21 |
| H2GCN | 59.39±1.98 | 37.90±2.02 | 35.86±1.03 | 82.16±4.80 | 84.86±6.77 | 86.67±4.69 |
| FAGCN | 63.44±2.05 | 41.17±1.94 | 36.81±0.26 | 81.35±5.05 | 84.32±6.02 | 83.33±2.01 |
| GPR-GNN | 61.58±2.24 | 46.65±1.81 | 35.27±1.04 | 81.89±5.93 | 83.24±4.95 | 84.12±3.45 |
| DeepWalk | 47.74±2.05 | 32.93±1.58 | 22.78±0.64 | 39.18±5.57 | 46.49±6.49 | 33.53±4.92 |
| node2vec | 41.93±3.29 | 22.84±0.72 | 28.28±1.27 | 42.94±7.46 | 41.92±7.76 | 37.45±7.09 |
| GAE | 33.84±2.77 | 28.03±1.61 | 28.03±1.18 | 58.85±3.21 | 58.64±4.53 | 52.55±3.80 |
| VGAE | 35.22±2.71 | 29.48±1.48 | 26.99±1.56 | 59.19±4.09 | 59.20±4.26 | 56.67±5.51 |
| DGI | 39.95±1.75 | 31.80±0.77 | 29.82±0.69 | 63.35±4.61 | 60.59±7.56 | 55.41±5.96 |
| GMI | 46.97±3.43 | 30.11±1.92 | 27.82±0.90 | 54.76±5.06 | 50.49±2.21 | 45.98±2.76 |
| MVGRL | 51.07±2.68 | 35.47±1.29 | 30.02±0.70 | 64.30±5.43 | 62.38±5.61 | 62.37±4.32 |
| GRACE | 48.05±1.81 | 31.33±1.22 | 29.01±0.78 | 54.86±6.95 | 57.57±5.68 | 50.00±5.83 |
| GRACE-FA | 52.68±2.14 | 35.97±1.20 | 32.55±1.28 | 67.57±4.98 | 64.05±7.46 | 63.73±6.81 |
| GCA | 49.80±1.81 | 35.50±0.91 | 29.65±1.47 | 55.41±4.56 | 59.46±6.16 | 50.78±4.06 |
| BGRL | 47.46±2.74 | 32.64±0.78 | 29.86±0.75 | 57.30±5.51 | 59.19±5.85 | 52.35±4.12 |
| GAUSS | **76.89±1.87** | **67.93±1.40** | **37.37±0.76** | **82.69±3.39** | **85.38±2.28** | **87.82±3.28** |

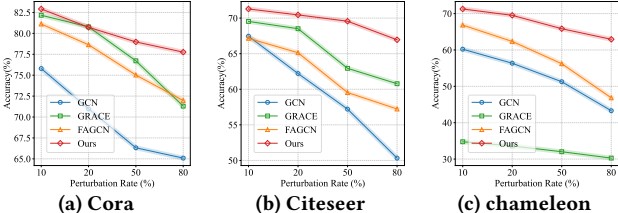

Figure 6: Performance with adding noisy edges.

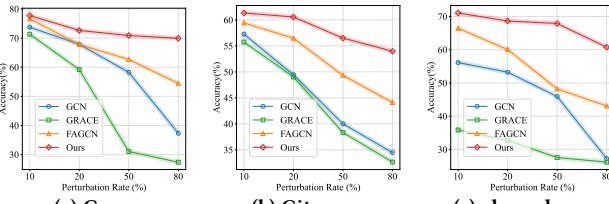

Figure 7: Performance with adding noisy attributes.

*5.2.2 Effectiveness analysis.* To illustrate the effectiveness of GAUSS, we plotted heatmaps on the synthetic and real world datasets, from left to right: The propagation matrix obtained by our proposed method GAUSS for each ego-network: matrix B, matrix Z, adjacency matrix and attention matrix, adjacency matrix, and attention matrix. Detailed information about the synthetic dataset csbm can be found in Appendix B, where $h$ stands for the degree of congruence. For consistency and ease of presentation, we normalize the B and Z matrices and add self-loops to the B and adjacency matric, and the different classes are marked with red lines.

From Figure 4 and Figure 5, it can be seen that no matter it is the synthetic dataset or the real world datasets, the matrices we get are better than the adjacency matrix and attention matrix, Both matrix B and matrix Z are classified into different blocks by node attribute similarity, which ensures that the attributes of similar nodes only propagate within blocks. At the same time, this approach overcomes the limitations of the original topology, and information can be exchanged between nodes of the same class that are otherwise unconnected, while the propagation of information between nodes of the different classes that are otherwise connected is greatly reduced, which is the superiority of our approach compared to the original topology-based approach.

*5.2.3 Robustness Analysis.* In this experiment, we investigate the robustness of GAUSS on graph data. This involves randomly adding noisy edges and attributes, respectively, followed by testing the accuracy of node classification on the learned representations from the perturbed graphs. We compare the classical GCN, the self-supervised method GRACE and the heterophilic graph method FAGCN on the Cora, Citeseer and Chameleon datasets.

From Figure 6 and Figure 7, it is clear that GAUSS consistently performs better than the baselines by a significant margin. Additionally, as the rate of perturbation increases, the superiority of our method becomes more pronounced, the performance of GCN and GRACE decreases significantly, FAGCN also showed some decreases, meaning that they are more sensitive to noise. While the performance of GAUSS performance is relatively stable, which shows the robustness of GAUSS, as shown in Figure 6. In the meantime, It also reports that GAUSS is also superior to other baselines under attributes perturbation, which can be attributed to the block diagonal representation in our method, which strictly limits the propagation of information between similar nodes while having some anti-noise effect. These experimental results demonstrate the strong robustness of GAUSS against random attacks on graph topology and node attributes. Figure 7 also reports that the performance degradation of GAUSS is slight and outperforms GCN, GRACE

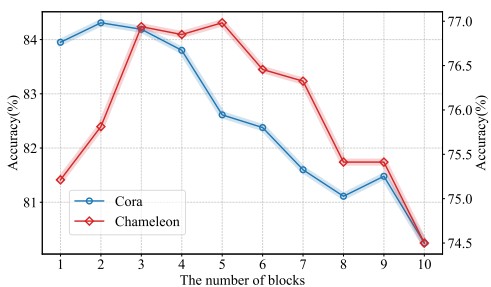

(a) $k = 2$     (b) $k = 3$

Figure 8: A local perspective on the number of blocks.

Figure 9: Performance with different number of blocks.

and FAGCN with different levels of noise interference, which can be attributed to the denoising in GAUSS may take the high-order relationships in the ego-nework. The experimental results demonstrate the strong robustness of GAUSS against adversarial attacks on graph struture.

## 5.3 Hyperparameter Analysis

In order to analyze the performance of our proposed module, we have experimented with some parameters of the module. Our aim is that for each ego network, we would like to classify nodes belonging to the same category into the same block and nodes of different categories into different blocks. However, as unsupervised learning, we are not supposed to know the number of classes in the dataset before experimenting, so this value was explored experimentally.

Figure 8 shows, as an example, the local variations when different values of k are taken, and we can see that from k = 2 to 3 there is a slight decrease in the weight of the two nodes propagating to each other inside class 0, with a tendency to split into two blocks.

As shown in Figure 9, we ran experiments on the homophilic dataset Cora and the heterophilic dataset Chameleon, respectively, which were used to explore the effect of different numbers of blocks in an ego network on node classification performance.For Cora, the performance reaches better results when the number of blocks is small and decreases as the number increases. This indicates that for the homophilic dataset, most of the ego-network has only one or two types of nodes in it. On the other hand, for Chameleon, the performance initially increases as the number of blocks increases and after a certain number the performance starts to decrease. This also confirms that there are multi-class nodes within the ego-network of the homophilic dataset.

For other parameters such as $\lambda$ and $\gamma$, we provide a detailed description in Appendix C.

## 6 VISUALIZATION

In order to provide a more illustrative perspective of the performance of our model, we have used t-SNE visualisation to provide a more intuitive view. Figure 10 shows t-SNE visualisations of

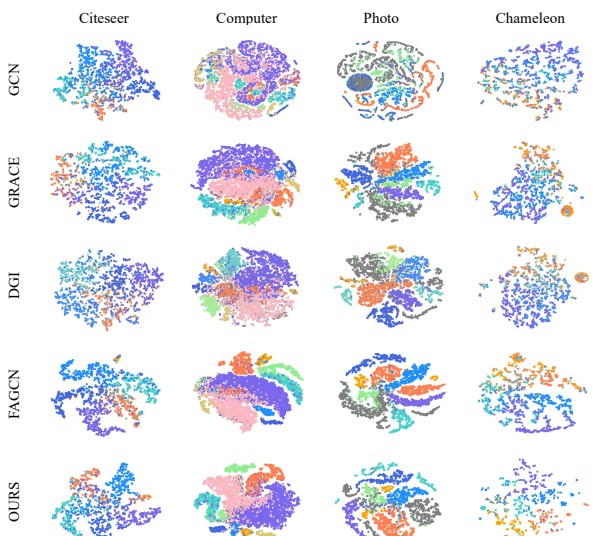

Figure 10: The visualization for node representations.

node embeddings obtained by GCN, GRACE, DGI, FAGCN and our method on four different datasets. We use different colours to represent different categories, showing the clustering effect of node embeddings. The shapes of these clusters reflect the characteristics of the respective models.

In particular, in the case of GCN, the embedding clusters for different categories tend to overlap, indicating a susceptibility to underfitting. Compared to the self-supervised methods GRACE and DGI, the clustering phenomenon of our methods is better on the homophilic graph dataset, which is particularly evident in the Citeseer dataset. At the same time, these two methods are significantly less effective in dividing on the heterophilic graph dataset Chameleon, which is also consistent with their poor performance. For FAGCN, our method also shows good competitiveness and there is less overlap between nodes of different colours on the heterogeneous graph. In contrast, the clusters produced by our method are more regular, and nodes with the same label tend to exhibit spatial clustering. This highlights the discriminative power of our method.

## 7 CONCLUSIONS

The universality and the generalization are two requirements of the Web. Existing Graph Neural Networks (GNNs) can separately meet them. Unfortunately, graph universal self-supervised learning (SSL) remains resolved. Most existing SSL can neither alter the low-passing filtering characteristic of GCN nor learn the graph via universal GNNs. To overcome this difficulty, this paper proposes novel GrAph-customized Universal Self-Supervised Learning (GAUSS) by exploiting local attribute distribution. The main idea is to replace the global parameters with locally learnable propagation. To make the propagation matrix demonstrate the affinity between the nodes, the self-representative learning framework is employed with $k$-block diagonal regularization. Extensive experiments on synthetic and real-world datasets demonstrate its effectiveness, universality, and robustness to noises.

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

## A  OPTIMIZATION OF GAUSS

We demonstrate a method for addressing the nonconvex problem defined in Eq. (10). The primary obstacle in this endeavor pertains to the nonconvex component $\|\mathbf{B}\|_{kb}$. To tackle this issue, we leverage a noteworthy property related to the sum of eigenvalues, as introduced by Ky Fan, to redefine our approach.

THEOREM A.1. *Let* $\mathbf{L} \in \mathbb{R}^{n \times n}$ *and* $\mathbf{L} \geq 0$. *Then* $\sum_{i=n-k+1}^{n} \lambda_i(\mathbf{L}) = \min_{\mathbf{W}} \langle \mathbf{L}, \mathbf{W} \rangle$, *s.t.* $0 \leq \mathbf{W} \leq \mathbf{I}, \mathrm{Tr}(\mathbf{W}) = k$. *Then, we can reformulate* $\|\mathbf{B}\|_{kb}$ *as a convex program* $\|\mathbf{B}\|_{kb} = \min_{\mathbf{W}} \langle \mathbf{L_B}, \mathbf{W} \rangle$, *s.t.* $0 \leq \mathbf{W} \leq \mathbf{I}$, $\mathrm{Tr}(\mathbf{W}) = k$.

So (10) is equivalent to

$$\min_{\mathbf{Z},\mathbf{B},\mathbf{W}} \frac{1}{2}\|\mathbf{X} - \mathbf{XZ}\|^2 + \frac{\lambda}{2}\|\mathbf{Z} - \mathbf{B}\|^2 + \gamma \langle \mathrm{Diag}(\mathbf{B1}) - \mathbf{B}, \mathbf{W}\rangle$$
$$\text{s.t. } \mathrm{diag}(\mathbf{B}) = 0, \mathbf{B} \geq 0, \mathbf{B} = \mathbf{B}^\top, \tag{12}$$
$$0 \leq \mathbf{W} \leq \mathbf{I}, \mathrm{Tr}(\mathbf{W}) = k.$$

In Eq. (12), it is evident that the problem comprises three distinct blocks of variables. We observe that W is independent from Z, thus we can group them as a super block {W, Z} and treat {B} as the other block. Then Eq. (12) can be solved by alternating updating {W, Z} and {B}.

First, fix $\mathbf{B} = \mathbf{B}^k$, and update $\{\mathbf{W}^{k+1}, \mathbf{Z}^{k+1}\}$ by

$$\{\mathbf{W}^{k+1}, \mathbf{Z}^{k+1}\} = \arg\min_{\mathbf{W},\mathbf{Z}} \frac{1}{2}\|\mathbf{X} - \mathbf{XZ}\|^2 + \frac{\lambda}{2}\|\mathbf{Z} - \mathbf{B}\|^2$$
$$+ \gamma \langle \mathrm{Diag}(\mathbf{B1}) - \mathbf{B}, \mathbf{W}\rangle \tag{13}$$
$$\text{s.t. } 0 \leq \mathbf{W} \leq \mathbf{I}, \mathrm{Tr}(\mathbf{W}) = k.$$

This is equivalent to updating $\mathbf{W}^{k+1}$ and $\mathbf{Z}^{k+1}$ separably by

$$\mathbf{W}^{k+1} = \arg\min_{\mathbf{W}} \langle \mathrm{Diag}(\mathbf{B1}) - \mathbf{B}, \mathbf{W}\rangle,$$
$$\text{s.t. } 0 \leq \mathbf{W} \leq \mathbf{I}, \mathrm{Tr}(\mathbf{W}) = k, \tag{14}$$

and

$$\mathbf{Z}^{k+1} = \arg\min_{\mathbf{Z}} \frac{1}{2}\|\mathbf{X} - \mathbf{XZ}\|^2 + \frac{\lambda}{2}\|\mathbf{Z} - \mathbf{B}\|^2. \tag{15}$$

Second, fix $\mathbf{W} = \mathbf{W}^{k+1}$ and $\mathbf{Z} = \mathbf{Z}^{k+1}$, and update B by

$$\mathbf{B}^{k+1} = \arg\min_{\mathbf{B}} \frac{\lambda}{2}\|\mathbf{Z} - \mathbf{B}\|^2 + \gamma \langle \mathrm{Diag}(\mathbf{B1}) - \mathbf{B}, \mathbf{W}\rangle$$
$$\text{s.t. } \mathrm{diag}(\mathbf{B}) = 0, \mathbf{B} \geq 0, \mathbf{B} = \mathbf{B}^\top. \tag{16}$$

Note that the aforementioned three subproblems are convex in nature and possess closed-form solutions. In Eq. (14), the update for $\mathbf{W}^{k+1}$ is given by $\mathbf{W}^{k+1} = \mathbf{U}\mathbf{U}^\top$, where $\mathbf{U} \in \mathbb{R}^{n \times k}$ comprises k eigenvectors associated with the k smallest eigenvalues of $\mathrm{Diag}(\mathbf{B1}) - \mathbf{B}$. As for Eq. (15), it is evident that

$$\mathbf{Z}^{k+1} = (\mathbf{X}^\top\mathbf{X} + \lambda\mathbf{I})^{-1}(\mathbf{X}^\top\mathbf{X} + \lambda\mathbf{B}). \tag{17}$$

For Eq. (16), it is equivalent to

$$\mathbf{B}^{k+1} = \arg\min_{\mathbf{B}} \frac{1}{2}\left\|\mathbf{B} - \mathbf{Z} + \frac{\gamma}{\lambda}\left(\mathrm{diag}(\mathbf{W})\mathbf{1}^\top - \mathbf{W}\right)\right\|^2$$
$$\text{s.t. } \mathrm{diag}(\mathbf{B}) = 0, \mathbf{B} \geq 0, \mathbf{B} = \mathbf{B}^\top. \tag{*}$$

This problem has a closed form solution given as follows.

---

**Algorithm 1:** Solve Eq. (10) by Alternating Minimization

**Data:** Initialize: $k = 0, W_k = 0, Z_k = 0, B_k = 0$
1 **while** *not converged* **do**
2     Compute $W_{k+1}$ by solving Eq. (14);
3     Compute $Z_{k+1}$ by solving Eq. (15);
4     Compute $B_{k+1}$ by solving Eq. (16);
5     $k = k + 1$;
6 **end**

---

PROPOSITION A.2. *Let* $\mathbf{A} \in \mathbb{R}^{n \times n}$. *Define* $\hat{\mathbf{A}} = \mathbf{A} - \mathrm{Diag}(\mathrm{diag}(\mathbf{A}))$. *Then the solution to the following problem*

$$\min_{\mathbf{B}} \frac{1}{2}\|\mathbf{B} - \mathbf{A}\|^2, \quad \text{s.t. } \mathrm{diag}(\mathbf{B}) = 0, \mathbf{B} \geq 0, \mathbf{B} = \mathbf{B}^\top, \tag{18}$$

*is given by* $\mathbf{B}^* = \left[(\hat{\mathbf{A}} + \hat{\mathbf{A}}^\top)/2\right]_+$.

The complete procedure of the alternating minimization solver for Eq. (10) is outlined in Algorithm 1. We represent the objective function of Eq. (10) as $f(\mathbf{Z}, \mathbf{B}, \mathbf{W})$. Let $S_1 = \{\mathbf{B}|\mathrm{diag}(\mathbf{B}) = 0, \mathbf{B} \geq \mathbf{B} = \mathbf{B}^\top\}$ and $S_2 = \{\mathbf{W}|0 \leq \mathbf{W} \leq \mathbf{I}, \mathrm{Tr}(\mathbf{W}) = k\}$. The indicator functions of $S_1$ and $S_2$ are denoted as $\iota_{S_1}(\mathbf{B})$ and $\iota_{S_2}(\mathbf{W})$, respectively. We provide the convergence guarantee for Algorithm 1 in the context of the nonconvex GAUSS problem.

PROPOSITION A.3. *The sequence* $\{\mathbf{W}^{k+1}, \mathbf{Z}^{k+1}, \mathbf{B}^{k+1}\}$ *generated by Algorithm 1 has the following properties:*

*(1) The objective* $f(\mathbf{Z}^k, \mathbf{B}^k, \mathbf{W}^k) + \iota_{S_1}(\mathbf{B}^k) + \iota_{S_2}(\mathbf{W}^k)$ *is monotonically decreasing. Indeed,*

$$f(\mathbf{Z}^{k+1}, \mathbf{B}^{k+1}, \mathbf{W}^{k+1}) + \iota_{S_1}(\mathbf{B}^{k+1}) + \iota_{S_2}(\mathbf{W}^{k+1})$$
$$\leq f(\mathbf{Z}^k, \mathbf{B}^k, \mathbf{W}^k) + \iota_{S_1}(\mathbf{B}^k) + \iota_{S_2}(\mathbf{W}^k)$$
$$- \frac{\lambda}{2}\left\|\mathbf{Z}^{k+1} - \mathbf{Z}^k\right\|^2 - \frac{\lambda}{2}\left\|\mathbf{B}^{k+1} - \mathbf{B}^k\right\|^2;$$

*(2)* $\mathbf{Z}^{k+1} - \mathbf{Z}^k \to 0, \mathbf{B}^{k+1} - \mathbf{B}^k \to 0$ *and* $\mathbf{W}^{k+1} - \mathbf{W}^k \to 0$;
*(3) The sequences* $\{\mathbf{Z}^k\}, \{\mathbf{B}^k\}$ *and* $\{\mathbf{W}^k\}$ *are bounded.*

THEOREM A.4. *The sequence* $\{\mathbf{W}^k, \mathbf{Z}^{k+}, \mathbf{B}^{k+1}\}$ *generated by Algorithm 1 has at least one limit point and any limit point* $(\mathbf{Z}^*, \mathbf{B}^*, \mathbf{W}^*)$ *of* $\{\mathbf{Z}^k, \mathbf{B}^k, \mathbf{W}^k\}$ *is a stationary point of Eq. (12).*

Please consult the supplementary material for the proof of the theorem above. In general, our proposed solver in Algorithm 1 for the nonconvex GAUSS model is straightforward. The practical convergence guarantee provided in Theorem A.4 for Algorithm 1 is achieved without relying on unverifiable assumptions.

### A.1  Proof of Proposition A.2

It is obvious that problem Eq. (18) is equivalent to

$$\min_{\mathbf{B}} \frac{1}{2}\|\mathbf{B} - \hat{\mathbf{A}}\|^2, \text{ s.t. } \mathbf{B} \geq 0, \mathbf{B} = \mathbf{B}^\top. \tag{19}$$

The constraint $\mathbf{B} = \mathbf{B}^\top$ implies that $\|\mathbf{B} - \hat{\mathbf{A}}\|^2 = \|\mathbf{B} - \hat{\mathbf{A}}^\top\|^2$. Consequently

$$\frac{1}{2}\|\mathbf{B} - \hat{\mathbf{A}}\|^2 = \frac{1}{4}\|\mathbf{B} - \hat{\mathbf{A}}\|^2 + \frac{1}{4}\|\mathbf{B} - \hat{\mathbf{A}}^\top\|^2$$
$$= \frac{1}{2}\left\|\mathbf{B} - \frac{\hat{\mathbf{A}} + \hat{\mathbf{A}}^\top}{2}\right\|^2 + c(\hat{\mathbf{A}}) \tag{20}$$

Since $c(\hat{\mathbf{A}})$ is solely dependent on $\hat{\mathbf{A}}$, it follows that Eq. (19) is synonymous with

$$\min_{\mathbf{B}} \frac{1}{2} \left\| \mathbf{B} - (\hat{\mathbf{A}} + \hat{\mathbf{A}}^{\top})/2 \right\|^2, \text{ s.t. } \mathbf{B} \geq 0, \mathbf{B} = \mathbf{B}^{\top}, \quad (21)$$

this has a solution of $\mathbf{B}^* = \left[ (\hat{\mathbf{A}} + \hat{\mathbf{A}}^{\top})/2 \right]_+$.

## A.2 Proof of Proposition A.3

To begin with, based on the optimality of $\mathbf{W}^{k+1}$ to Eq. (14), we can deduce

$$f(\mathbf{Z}^k, \mathbf{B}^k, \mathbf{W}^{k+1}) + \iota_{S_2}(\mathbf{W}^{k+1}) \leq f(\mathbf{Z}^k, \mathbf{B}^k, \mathbf{W}^k) + \iota_{S_2}(\mathbf{W}^k). \quad (22)$$

Secondly, by considering the updating rule of $\mathbf{Z}^{k+1}$ in Eq. (15), we have

$$\mathbf{Z}^{k+1} = \arg\min_{\mathbf{Z}} f(\mathbf{Z}, \mathbf{B}^k, \mathbf{W}^{k+1}). \quad (23)$$

Note that $f(\mathbf{Z}^{k+1}, \mathbf{B}^k, \mathbf{W}^{k+1})$ is $\lambda$-strongly convex. This implies

$$f(\mathbf{Z}^{k+1}, \mathbf{B}^k, \mathbf{W}^{k+1}) \leq f(\mathbf{Z}^k, \mathbf{B}^k, \mathbf{W}^{k+1}) - \frac{\lambda}{2} \left\| \mathbf{Z}^{k+1} - \mathbf{Z}^k \right\|^2, \quad (24)$$

Furthermore, from the update rule of $\mathbf{B}^{k+1}$ in Eq. (16), we can observe that

$$\mathbf{B}^{k+1} = \arg\min_{\mathbf{B}} f(\mathbf{Z}^{k+1}, \mathbf{B}, \mathbf{W}^{k+1}) + \iota_{S_1}(\mathbf{B}). \quad (25)$$

Take into account that $f(\mathbf{Z}^{k+1}, \mathbf{B}, \mathbf{W}^{k+1}) + \iota_{S_1}(\mathbf{B})$ is $\lambda$-strongly convex w.r.t. $\mathbf{B}$.

$$\begin{aligned}
&f(\mathbf{Z}^{k+1}, \mathbf{B}^{k+1}, \mathbf{W}^{k+1}) + \iota_{S_1}(\mathbf{B}^{k+1}) \\
&\leq f(\mathbf{Z}^{k+1}, \mathbf{B}^k, \mathbf{W}^{k+1}) + \iota_{S_1}(\mathbf{B}^k) - \frac{\lambda}{2} \left\| \mathbf{B}^{k+1} - \mathbf{B}^k \right\|^2.
\end{aligned} \quad (26)$$

By combining the preceding equations, we obtain

$$\begin{aligned}
&f(\mathbf{Z}^{k+1}, \mathbf{B}^{k+1}, \mathbf{W}^{k+1}) + \iota_{S_1}(\mathbf{B}^{k+1}) + \iota_{S_2}(\mathbf{W}^{k+1}) \\
&\leq f(\mathbf{Z}^k, \mathbf{B}^k, \mathbf{W}^k) + \iota_{S_1}(\mathbf{B}^k) + \iota_{S_2}(\mathbf{W}^k) \\
&\quad - \frac{\lambda}{2} \left\| \mathbf{B}^{k+1} - \mathbf{B}^k \right\|^2 - \frac{\lambda}{2} \left\| \mathbf{Z}^{k+1} - \mathbf{Z}^k \right\|^2.
\end{aligned} \quad (27)$$

Therefore, $f(\mathbf{Z}^k, \mathbf{B}^k, \mathbf{W}^k) + \iota_{S_1}(\mathbf{B}^k) + \iota_{S_2}(\mathbf{W}^k)$, is steadily decreasing and, as a result, it is limited from above. Consequently, we can conclude that both $\{\mathbf{Z}^k\}$ and $\{\mathbf{B}^k\}$ are confined. Additionally, since $\mathbf{W}^k \in S_2$, it implies that $\left\| \mathbf{W}^k \right\|_2 \leq 1$, which in turn indicates that $\{\mathbf{W}^k\}$ is also bounded.

Please note that $\mathbf{W}^k$ and $\text{Diag}(\mathbf{B}^k \mathbf{1}) - \mathbf{B}^k$ are positive semidefinite. Consequently, we can establish $\langle \text{Diag}(\mathbf{B}^k \mathbf{1}) - \mathbf{B}^k, \mathbf{W}^k \rangle \geq 0$. As a result, $f(\mathbf{Z}^k, \mathbf{B}^k, \mathbf{W}^k) + \iota_{S_1}(\mathbf{B}^k) + \iota_{S_2}(\mathbf{W}^k) \geq 0$. Now, summing Eq. (27) over $k = 0, 1, \cdots$ we obtain the following

$$\sum_{k=0}^{+\infty} \frac{\lambda}{2} \left( \left\| \mathbf{B}^{k+1} - \mathbf{B}^k \right\|^2 + \left\| \mathbf{Z}^{k+1} - \mathbf{Z}^k \right\|^2 \right) \leq f(\mathbf{Z}^0, \mathbf{B}^0, \mathbf{W}^0). \quad (28)$$

This suggests

$$\mathbf{B}^{k+1} - \mathbf{B}^k \to 0, \quad (29)$$

And

$$\mathbf{Z}^{k+1} - \mathbf{Z}^k \to 0, \quad (30)$$

And the updating of $\mathbf{W}^{k+1}$, we have

$$\mathbf{W}^{k+1} - \mathbf{W}^k \to 0, \quad (31)$$

The proof is finished.

## A.3 Proof of Theorem A.4

Now, from the boundedness of $\{\mathbf{Z}^k, \mathbf{B}^k, \mathbf{W}^k\}$, there exists a point $(\mathbf{Z}^*, \mathbf{B}^*, \mathbf{W}^*)$ and a subsequence $\{\mathbf{Z}^{k_j}, \mathbf{B}^{k_j}, \mathbf{W}^{k_j}\}$, such that $\mathbf{Z}^{k_j} \to \mathbf{Z}^*, \mathbf{B}^{k_j} \to \mathbf{B}^*$, and $\mathbf{W}^{k_j} \to \mathbf{W}^*$. Then, we have $\mathbf{Z}^{k_j+1} \to \mathbf{Z}^*, \mathbf{B}^{k_j+1} \to \mathbf{B}^*$ and $\mathbf{W}^{k_j+1} \to \mathbf{W}^*$. On the other hand, from the optimality of $\mathbf{W}^{k_j+1}$ to Eq. (14), $\mathbf{Z}^{k_j+1}$ to Eq. (15) and $\mathbf{B}^{k_j+1}$ to Eq. (16), we have

$$0 \in \nabla f_{\mathbf{W}}(\mathbf{Z}^{k_j}, \mathbf{B}^{k_j}, \mathbf{W}^{k_j+1}) + \partial_{\mathbf{W}} \iota_{S_2}(\mathbf{W}^{k_j+1}),$$

$$0 \in \nabla f_{\mathbf{Z}}(\mathbf{Z}^{k_j+1}, \mathbf{B}^{k_j}, \mathbf{W}^{k_j+1}),$$

$$0 \in \nabla f_{\mathbf{B}}(\mathbf{Z}^{k_j+1}, \mathbf{B}^{k_j+1}, \mathbf{W}^{k_j+1}) + \partial_{\mathbf{B}} \iota_{S_1}(\mathbf{B}^{k_j+1})$$

Let $k \to +\infty$. We have

$$0 \in \nabla f_{\mathbf{W}}(\mathbf{Z}^*, \mathbf{B}^*, \mathbf{W}^*) + \partial_{\mathbf{W}} \iota_{S_2}(\mathbf{W}^*),$$

$$0 \in \nabla f_{\mathbf{Z}}(\mathbf{Z}^*, \mathbf{B}^*, \mathbf{W}^*),$$

$$0 \in \nabla f_{\mathbf{B}}(\mathbf{Z}^*, \mathbf{B}^*, \mathbf{W}^*) + \partial_{\mathbf{B}} \iota_{S_1}(\mathbf{B}^*).$$

Thus $(\mathbf{Z}^*, \mathbf{B}^*, \mathbf{W}^*)$ is a stationary point of Eq. (12)

## B SYNTHETIC DATASETS CONTEXTUAL STOCHASTIC BLOCK MODEL(CSBM)

The synthetic datasets cSBM allows for smoothly controlling the "informativeness ratio" between node features and graph topology, where the graph can vary from being highly homophilic to highly heterophilic. We consider the case with two equal-size classes. In cSBMs, the node features are Gaussian random vectors, where the mean of the Gaussian depends on the community assignment. The difference of the means is controlled by a parameter $\mu$, while the difference of the edge densities in the communities and between the communities is controlled by a parameter $\lambda$. Hence $\mu$ and $\lambda$ capture the "relative informativeness" of node features and the graph topology, respectively.

The cSBM adds Gaussian random vectors as node features on top of the classical SBM. For simplicity, we assume $C = 2$ equally sized communities with node labels vi in $\{+1, -1\}$. Each node $i$ is associate with a $f$ dimensional Gaussian vector $b_i = \sqrt{\frac{\mu}{n}} v_i u + \frac{Z_i}{\sqrt{f}}$, where n is the number of nodes, $u \sim N(0, I/f)$ and $Z_i \in \mathbf{R}^f$ has independent standard normal entries. The (undirected) graph in cSBM is described by the adjacency matrix $\mathbf{A}$ defined as

$$\mathbf{P}(\mathbf{A}_{ij} = 1) = \begin{cases} \frac{d + \lambda\sqrt{d}}{n} & \text{if } v_i v_j > 0 \\ \frac{d - \lambda\sqrt{d}}{n} & \text{otherwise} \end{cases} \quad (32)$$

Similar to the classical SBM, given the node labels the edges are independent. The symbol d stands for the average degree of the graph. Also, recall that $\mu$ and $\lambda$ control the information strength carried by the node features and the graph structure respectively.

## C PARAMETERS

In this section, based on cSBM dataset, we give a visualisation of the two parameters $\lambda$ and $\gamma$ during the iteration of the algorithm. The parameter $\lambda$ mainly constrains the degree of similarity between matrix Z and matrix B. Figure 11 shows the local variations when $\lambda$ is taken as 1, 10, and 100 respectively, and it can be seen that the differentiation is not very effective when $\lambda$ is taken too large or too small.

Figure 12 shows the local variations when $\gamma$ is taken as 0.1, 1, 10 and 50 respectively. The parameter $\gamma$ mainly affects the strength of the block-forming constraints, when $\gamma$ is too weak, it results in an increase in the inter-class propagation coefficient, and when it's too strong, it reduces the intra-class weight.

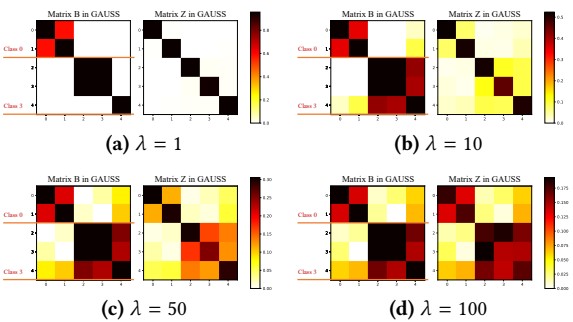

Figure 11: A local perspective on different $\lambda$.

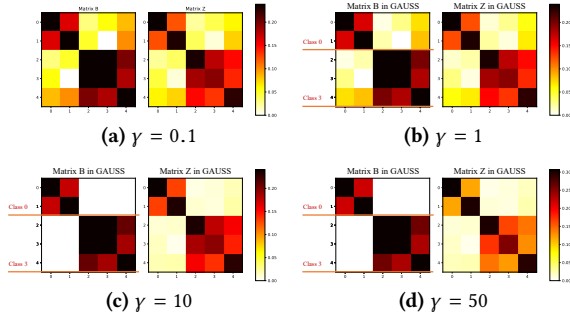

Figure 12: A local perspective on different $\gamma$.

