# OpenReview forum: "GAUSS: GrAph-customized Universal Self-Supervised Learning"
_ACM.org/TheWebConf/2024/Conference — TheWebConf24_

### Official Review · Reviewer_RXPg · 2023-11-19

**Novelty:** 4
**Technical Quality:** 4

**Review:**

The paper is motivated by the observation that SSL must be customized for the graph but universal GNNs is not fully taken advantage of in current SSL methods. Therefore, the authors propose GAUSS to exploit local attribute distribution.

pros:
1. The paper is well-written.
2. Good theoretical understanding and empirical results.

cons:
1. Weak experimental evaluations. The authors claim that they aim to "to make GNN meet the requirements of the Web," but only evaluates the proposed method on small graphs. The largest graph evaluted has less than 20k nodes, which is very small. Real-world web-scale graphs contain millions (if not billions) of nodes and edges, such as the ones in OGBN.
2. Organization can be improved. The paper places more emphasis on preliminaries and analysis (motivation) than methodology itself, making the methodology too concise and difficult to follow.
3. Some claims are unclear or unexplained. It is reasonable that " the expressive power of flexible GNN encoders can’t be fully exploited by self-supervised learning", but I do not understand how the authors' proposal is able to take advantage of the more flexible GNN encoders better than existing approaches. Moreover, it is unclear how avoiding "the employment of GCN" has to do with "customizing SSL for graph".

**Questions:**

Please see cons section in review.

**Reviewer Confidence:**

3: The reviewer is confident but not certain that the evaluation is correct

**Scope:**

3: The work is somewhat relevant to the Web and to the track, and is of narrow interest to a sub-community

---

### Official Review · Reviewer_MAs1 · 2023-11-27

**Novelty:** 6
**Technical Quality:** 6

**Review:**

Strengths: The paper presents a self-supervised method for graphs. The experiments showed effectiveness.

Weaknesses:
1. The proposed method is just an application or trivial extension of the self-expressive model widely used in subspace clustering [1]. The novelty is very limited. The regularizer $\Vert\mathbf{B}_ i\Vert_ {k b}$ is not new. It has been used by a few papers such as [2].
[1] Elhamifar and Vidal. Sparse Subspace Clustering: Algorithm, Theory, and Applications. IEEE TPAMI 2013
[2] Sun et al. Laplacian-based Cluster-Contractive t-SNE for High-Dimensional Data Visualization. TKDD 2023.
2. The optimization details for Problem (10) are missing.
3. The writing is poor. For instance, it is not clear why there is always an index $i$ in (6)(7)(10)(11).
4. The improvement over the baselines seems tiny, as shown by Table 2.

**Questions:**

Please refer to the review session.

**Reviewer Confidence:**

4: The reviewer is certain that the evaluation is correct and very familiar with the relevant literature

**Scope:**

4: The work is relevant to the Web and to the track, and is of broad interest to the community

---

### Official Review · Reviewer_KKub · 2023-11-28

**Novelty:** 4
**Technical Quality:** 4

**Review:**

**Summary**

The paper proposes a universal graph self-supervised learning framework that aims to achieve good performance on both of the homophilic and heterophilic graphs. Theoretical analyses are provided to justify the necessity of the work and comprehensive experiments are conducted to demonstrate the effectiveness of the proposed framework.

**Strengths**
1. The problem that the paper studies is important and of great practical impact.
2. The motivation of the paper is well justified by empirical and experimental studies.
3. The proposed framework is evaluated on multiple benchmarks, including homophilic, heterophilic, and synthetic datasets, against many existing baselines. The experimental results demonstrate the effectiveness of the proposed method.

**Weaknesses**

1.	Though the proposed framework is designed to address the issues of existing SSL graph representation learning methods, it is still essentially a graph SSL framework if my understanding is correct. In this case, it would be better if the authors could add a section to clarify the implementation details of the proposed method, like how the input graph is augmented and what kind of graph SSL objective is used.
2.	The authors should analyze the computation complexity of the proposed method and compare it against the baselines.
3.	The reported results of the baselines seem to be different from the original paper, the authors are encouraged to clarify the reason for the gap.

**Questions:**

Please refer the weakness section.

**Reviewer Confidence:**

4: The reviewer is certain that the evaluation is correct and very familiar with the relevant literature

**Scope:**

4: The work is relevant to the Web and to the track, and is of broad interest to the community

---

### Official Review · Reviewer_hBUG · 2023-11-30

**Novelty:** 5
**Technical Quality:** 5

**Review:**

The paper proposes a novel self-supervised learning framework called GAUSS for graph neural networks to achieve universality and generalization. GAUSS replaces global parameters with locally learnable propagation matrices customized for each node's ego-network. It employs a self-representative learning objective regularized with k-block diagonal constraints to learn affine propagation matrices that capture node affinity. Experiments show GAUSS outperforms GNN baselines on node classification across both homophilic and heterophilic graphs.

Strengths:
- Addresses an important open problem of achieving universal self-supervised learning on graphs
- Proposes a principled approach to learning graph structure locally in a self-supervised manner
- Achieves SOTA results on both homophilic and heterophilic graph datasets
- Demonstrates robustness against graph perturbations

Weaknesses:
- Effects of different k values for block regularization not fully analyzed
- Some notations are used multiple times, such as k (number of layers/blocks)

**Questions:**

1. How does runtime scale with number of ego-networks and their sizes?
2. Would an end-to-end trained model work better than the proposed alternating optimization?

**Reviewer Confidence:**

3: The reviewer is confident but not certain that the evaluation is correct

**Scope:**

4: The work is relevant to the Web and to the track, and is of broad interest to the community

---

### Decision · Program_Chairs · 2024-01-22

**Decision:**

Accept

**Comment:**

Pros:
 * The paper proposed a universal self-supervised learning method for both homophilic graphs and heterophilic graphs. The method shows outperforms a large collection of baseline methods (including semi-supervised methods and self-supervised methods) on a large collection of graphs. The performance improvement on heterophilic graphs is significant.
 * The motivation of the work is well justified in the paper.
 * The method is principled and novel.
 * The method is scalable (based on the computation complexity in the rebuttal) and the authors also demonstrated its scalability of relatively large graphs, such as ogbn-products, as well as SOTA model performance on these large graphs.

 Cons:
 * The writing can be improved. I noticed that there were multiple misunderstandings in the reviews and it took me a while to understand the proposed method. Essentially, the method is quite different from previous self-supervised methods. The authors should include many of the discussions in the paper.